

# Linearized regime of
# the generalized hydrodynamics with diffusion

## Miłosz Panfil⋆ and Jacek Pawełczyk†

Faculty of Physics, University of Warsaw, Pasteura 5, 02-093 Warsaw, Poland

⋆ milosz.panfil@fuw.edu.pl, † jacek.pawelczyk@fuw.edu.pl

## Abstract

We consider the generalized hydrodynamics including the recently introduced diffusion term for an initially inhomogeneous state in the Lieb-Liniger model. We construct a general solution to the linearized hydrodynamics equation in terms of the eigenstates of the evolution operator and study two prototypical classes of initial states: delocalized and localized spatially. We exhibit some general features of the resulting dynamics, among them, we highlight the difference between the ballistic and diffusive evolution. The first one governs a spatial scrambling, the second, a scrambling of the quasi-particles content. We also go one step beyond the linear regime and discuss the evolution of the zero momentum mode that does not evolve in the linear regime.

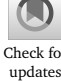
# 1  Introduction

The questions of dynamics and equilibration of an initially inhomogeneous state in integrable theories have been an area of active research. One common setup is the so called bi-partite quench protocol which refers to non-equilibrium dynamics of two macroscopically different subsystems joined together. Such bi-partite quenches or similar inhomogeneous initial states were studied in different contexts ranging from the CFT [1–7] and more generally QFT [8–10] to lattice models including 1d spin chains [11–22] and 1d Hubbard-like models [23–25]. The hydrodynamics solution to bi-partite quench protocol was originally proposed in [26,27] and further developed [28–34]. The resulting theory, Generalized Hydrodynamics (GHD), was successfully applied to variants of bi-partite quench protocols [27,35–38] and other inhomogeneous setups [28,39]. Its predictions were also confirmed experimentally [40]. Recently, based on the microscopic insights, diffusion was incorporated to the GHD picture [41–43].

In this work, we consider the resulting dynamics for the Lieb-Liniger model, a gas of 1d bosons with ultra-local interactions. The GHD forms a complicated, infinite, set of non-linear equations. Our aim is to analyze solutions to those equations and display some of their qualitative features, especially those relevant to the question of equilibration. Specifically, we focus on the linearized regime of the GHD in which the resulting equations take a form of an infinite set of integro-differential equations.

The manuscript is organized in the following way. After the introductory part of the next section, follow three sections with results. In Section 3 we display a general solution to the linear GHD in terms of the eigenstates of the diffusion operator. We also study numerically the properties of this operator and conjecture their effect on the dynamics. In the following Section 4 we solve numerically the GHD equations for two prototypical initial states: the localized and the delocalized state. In Section 5 we consider a next order correction to the linear GHD and study the dynamics of the $k = 0$ spatial mode which does not evolve in the linear approximation. Finally, the Appendix A is devoted to set the notation for integral operators.

# 2  Generalized hydrodynamics, diffusion and the Lieb-Liniger model

We start this section by presenting the Generalized Hydrodynamics following the original works [41,43]. Then, we recall the Lieb-Liniger model and introduce the ingredients necessary for its GHD description. We will consider the Lieb-Liniger model with repulsive interactions, which contains quasi-particles of only one type. Below, we present the GHD appropriate for this case. Generalization to situations with multi-species quasi-particles is possible [43].

## 2.1  Generalized hydrodynamics

The integrable theories are characterized by stable quasi-particles. The stability of the quasi-particles is related to the presence of an extensive number of local (or quasi-local) conserved densities $\hat{q}_i(x, t)$. The hydrodynamic picture relies on the assumption that the state of the system can be locally described by the averages $\bar{q}_i(x, t) = \langle \hat{q}_i(x, t) \rangle$. These averages can be conveniently parametrized introducing local distribution of the quasi-particles $\rho_p(\theta, t, x)$

$$\bar{q}_i(x, t) = \int d\theta \, \rho_p(\theta, x, t) h_i(\theta), \tag{1}$$

where $h_i(\theta)$ is the one-particle eigenvalue of the conserved charge with density $q_i(x, t)$ and rapidity $\theta$ parametrizes the quasi-particle. At the Euler scale $\rho_p(\theta, x, t)$ corresponds to the density of a local Generalized Gibbs Ensemble [44–46]. Beyond that scale, including the next

term in the hydrodynamic derivative expansion, the average values of generic observables depend not only on $\rho_p(\theta, x, t)$ but also on its spatial derivatives.

At the Navier-Stokes level the dynamics of the GHD can be understood through the following qualitative picture. When a quasi-particle travels through an inhomogeneous system it experiences two effects. First, the changing distribution of the quasi-particles leads to a difference in the propagation velocity. Second, the scattering processes with other particles lead to diffusion, see Fig. 2. These two effects are combined in Navier-Stokes-like equations of GHD

$$\partial_t \rho_p + \partial_x(v_n^{\text{eff}} \rho_p) = \frac{1}{2}\partial_x\left(\mathbf{D}_n \partial_x \rho_p\right), \tag{2}$$

with the effective velocities $v_n^{\text{eff}}$ and diffusion operator $\mathbf{D}_n$ depending on the local (in the space-time) distribution of quasi-particles. Solving (2) is difficult even in the absence of the diffusion term, however, in some cases the problem can be rephrased into more trackable integral equations [31]. The difficulty lies in the fact that each mode $\theta$ propagates with a velocity $v_n^{\text{eff}}(\theta)$ depending non-linearly on the local density function $\rho_p(\theta, t, x)$ of quasi-particles. Therefore, already at the ballistic level, the dynamics is complicated and non-linear [47] and displays the equilibration phenomena. The known results describe, for example, the self-similar solution [27] or numerical solution [38] for the nonequilibrium steady states.

Inclusion of the diffusion term in (2) further complicates the situation, e.g. it leads to mixing between densities of different rapidities $\theta$. Our aim is to understand better the role of the diffusion term in the dynamics governed by (2).

Let us specify now in some details the ingredients entering Eq. (2). The effective velocity is defined as

$$v_n^{\text{eff}}(\theta) = \frac{(E')^{\text{dr}}(\theta)}{(p')^{\text{dr}}(\theta)}. \tag{3}$$

The functions $(E')^{\text{dr}}(\theta)$ and $(p')^{\text{dr}}(\theta)$ are dressed derivatives of the effective energy and momentum of a particle with rapidity $\theta$ in the presence of the other particles specified by filling function $n(\theta)$ (see Appendix A for the definition of the dressing procedure and notation).

The diffusion operator is given by the integral kernel

$$\mathbf{D}_n = (1 - n\mathbf{T})^{-1} \rho_s \tilde{\mathbf{D}}_n \rho_s^{-1}(1 - n\mathbf{T}), \tag{4}$$

where $\mathbf{T}$ is the differential scattering kernel (explicitly defined for the Lieb-Liniger model in Sec. 2.2) and $\tilde{\mathbf{D}}_n$ is the diffusion kernel

$$\tilde{\mathbf{D}}_n(\theta, \alpha) = \frac{1}{\rho_s^2(\theta)}\left(\delta(\theta - \alpha)w(\alpha) - W(\theta, \alpha)\right). \tag{5}$$

Function $\rho_s(\theta)$ specifies the maximal allowed density of particles in the range $[\theta, \theta + \Delta\theta]$ and is related to the particles density $\rho_p(\theta)$ through the filling function $n(\theta)$: $\rho_s(\theta) = n(\theta)\rho_p(\theta)$. The other two function entering the formula above are

$$W(\theta, \alpha) = \rho_p(\theta)(1 - n(\theta))\left[T^{\text{dr}}(\theta, \alpha)\right]^2 |v_n^{\text{eff}}(\theta) - v_n^{\text{eff}}(\alpha)|, \tag{6}$$

$$w(\theta) = \int d\alpha\, W(\alpha, \theta). \tag{7}$$

The dependence of all the above functions on space-time variables has been skipped for the compactness of the notation.

## 2.2 Lieb-Liniger model

The Lieb-Liniger model [48, 49] describes a gas of bosonic particles in one spatial dimension with ultra-local interactions. The Hamiltonian for $N$ particles is

$$H_{\text{LL}} = -\sum_{j=1}^{N} \partial_x^2 + 2c \sum_{i<j} \delta(x_i - x_j), \tag{8}$$

where we set $\hbar = 2m = 1$. We focus on repulsive interactions, $c > 0$, and assume periodic boundary conditions for a system of length $L$. The Lieb-Liniger model is exactly solvable. At the center of the exact solution is the scattering kernel

$$T(\theta, \theta') = \frac{1}{\pi} \frac{c}{c^2 + (\theta - \theta')^2}. \tag{9}$$

The thermodynamics in the limit of finite particles density $N/L$ (which we assume from now on to be 1) is known [45, 50–52] and takes a universal form

$$n(\theta) = \frac{1}{1 + e^{\epsilon(\theta)}}, \tag{10}$$

where $\epsilon(\theta)$ solves an Thermodynamic Bethe Ansatz (TBA) equation. At the thermal equilibrium

$$\epsilon(\theta) = \frac{\theta^2 - \mu}{T} - \int_{-\infty}^{\infty} d\theta' \, T(\theta, \theta') \log\left(1 + e^{-\epsilon(\theta')}\right), \tag{11}$$

where $T$ is the temperature and $\mu$ is the chemical potential. The generalized TBA [45] describes in the similar framework the filling functions of non-thermal states. An example of such a state is a state reached after the interaction quench from the BEC state, the ground state of the non-interacting gas $c = 0$. In that case the distribution of quasi-particles is known explicitly to be [53]

$$n_{\text{quench}}(\theta) = \frac{a(\theta/c)}{a(\theta/c) + 1}, \qquad a(x) = \frac{2\pi}{cx \sinh(2\pi x)} I_{1-2ix}(4/\sqrt{c}) I_{1+2ix}(4/\sqrt{c}), \tag{12}$$

with $I_j(x)$ the modified Bessel functions of the first kind.

The bare momentum and energy are $p(\theta) = \theta$, $E(\theta) = \theta^2$. The physical observables in the Lieb-Liniger model depend on the particles density $\rho_p(\theta)$ which follows from the filling function $n(\theta)$ through the same dressing procedure

$$\rho_p = n(1 - \mathbf{T}n)^{-1} \frac{1}{2\pi}, \tag{13}$$

where we understand that $1/(2\pi)$ is a constant function.

In this work, we focus our attention on the filling function $n(\theta)$ instead of a particle density $\rho_p(\theta)$. The reason is two-fold. First, the linearized hydrodynamics is most easily formulated for the $n(\theta)$ function. Therefore to understand the resulting dynamics it is most natural to look at $n(\theta)$ instead of $\rho_p(\theta)$. Second, beside the non-linear relation the quantitative features of $\rho_p(\theta)$ are visible already at the level of $n(\theta)$. The main effect in the transformation (13) from $n(\theta)$ to $\rho_p(\theta)$ is in smoothing the shape of the function and, for small values of $c$, making the distribution more compact, see Fig. 1.

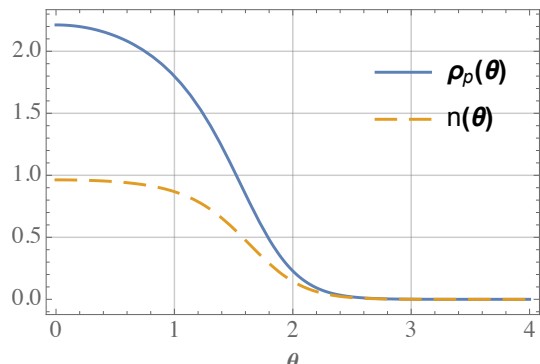

Figure 1: The particle density $\rho_p(\theta)$ and filling function $n(\theta)$ for a thermal gas at $T = 1$ and with $c = 1$.

## 3   Linearized generalized hydrodynamics

We consider now a linear approximation in perturbation $\delta n$ around space-time independent equilibrium $\underline{n}(\theta)$:

$$n(\theta, t, x) = \underline{n}(\theta) + \delta n(\theta, t, x). \tag{14}$$

This has two applications. First, when one indeed considers perturbing a space-time independent equilibrium with a small disturbance. [1] The final state of the GHD evolution is then the original equilibrium state.

The second case is in the large-time hydrodynamics of an initially strongly inhomogeneous state. We expect that at late times the GHD dynamics smooth out the quantum fluid to a state which can be described by the linearized equations around a homogeneous state. One must, however, remember that the hydrodynamical modes can only describe systems which are in local thermodynamical equilibrium [26].

The Eq. (2) of the GHD was expressed in terms of the particles' density $\rho_p(\theta, t, x)$. To consider its linear regime it is convenient to write it as an equation for the filling function $n(\theta, t, x)$. The equation is then [43]

$$\partial_t n + v_n^{\text{eff}} \partial_x n = \frac{1}{2\rho_s}(1 - n\mathbf{T})\partial_x \left((1 - n\mathbf{T})^{-1} \rho_s \tilde{\mathbf{D}}_n \partial_x n\right), \tag{15}$$

and its linearized form is

$$\partial_t \delta n(\theta, t, x) + v_{\underline{n}}^{\text{eff}} \partial_x \delta n(\theta, t, x) = \frac{1}{2} \int d\alpha \, \tilde{\mathbf{D}}_{\underline{n}}(\theta, \alpha) \partial_x^2 \delta n(\alpha, t, x). \tag{16}$$

Here we have explicitly written the action of the kernel operator on the filling function. The effective velocities $v^{\text{eff}}$ and the diffusion kernel $\tilde{\mathbf{D}}(\theta, \alpha)$ are determined from the equilibrium data $\underline{n}$ as in Eq.(14) [2].

The time evolution of linearized hydrodynamics (16) is driven by two phenomena. First, the velocity with which particles propagate depends on their rapidity. This leads to spatial spreading of an initially localized profile $\delta n(\theta, t, x = 0)$ even in the absence of the diffusion. On top of this the diffusion processes lead to the redistribution of the particle content $\delta n(\theta, t, x)$ through scattering processes with the background, see Fig. 2

The effect of the GHD dynamics is the reorganization of the spatial and rapidity dependence of the filling function. The linear dynamics is conservative in the sense that the total profile of

---

[1] For infinite systems $\delta n$ has to be additionally compactly supported.

[2] Dependence on $\underline{n}$ will be suppressed in our notation from now on.

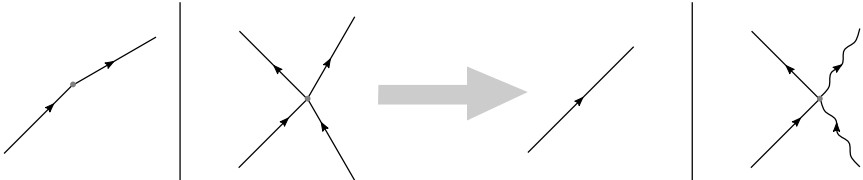

Figure 2: Left panel: The hydrodynamics is driven by two processes. First, is the ballistic propagation of particles with the velocity that depends on the local distribution of particles. The second process, that leads to the diffusion, is the scattering between the particles. Right panel: In the linearized regime both processes gets simplified. The ballistic propagation occurs now with the space-time constant velocity determined by the particles rapidity and the background state $\underline{n}(\theta)$. The scattering events, driving the diffusion, are now between the perturbation and the background particles and are controlled by the dressed scattering phase shift $T^{\mathrm{dr}}$.

rapidities

$$\int \mathrm{d}x\, \delta n(\theta, t, x), \tag{17}$$

does not vary in time.

## 3.1 Solution to the linearized GHD

Equation (16) is an integro-differential equation for $\delta n(\theta, t, x)$. The variables are separated and we can solve it in a standard manner. For a single momentum mode

$$\delta n(\theta, t, x) = \delta n_k(\theta, t) e^{ikx}, \tag{18}$$

(16) yields

$$\partial_t \delta n_k(\theta, t) + \int d\alpha\, \mathfrak{D}_k(\theta, \alpha) \delta n_k(\alpha, t) = 0\,, \tag{19}$$

where the operator

$$\mathfrak{D}_k(\theta, \alpha) = ikv^{\mathrm{eff}}(\theta)\delta(\theta - \alpha) + \frac{k^2}{2}\tilde{\mathbf{D}}(\theta, \alpha)\,, \tag{20}$$

explicitly depends on the Fourier mode $k$. Hereafter, we shall consider $k \neq 0$ modes only: $k = 0$ does not evolve in time at the linear level and should be part of the background equilibrium state. At the quadratic level $\delta n_0$ acquires non trivial dynamics as it shall be discussed in Sec. 5.

The operator $\mathfrak{D}_k$ being an integral, linear operator on non-compact quasi-momenta space has an infinite number of eigenstates $f_{k,\omega}(\theta)$ and corresponding eigenvalues $z_{k,\omega} \in \mathbb{C}$,

$$z_{k,\omega} \equiv i\kappa_\omega(k) + \lambda_\omega(k), \tag{21}$$

where $\omega$ enumerates the eigenvalues, $\kappa_\omega(k)$ and $\lambda_\omega(k)$ are both real and control the ballistic and diffusive propagation respectively. In writing (21) we have anticipated that, once for each $k$ a suitable ordering of $\omega$'s is made, the $z_{k,\omega}$'s are well-defined functions of $k$. From the numerical diagonalization, we observe that the spectrum of $\mathfrak{D}_k$ is non-degenerate and can be ordered with $\kappa_\omega$. An example of such spectrum is given in Fig. 3. From (20) it follows that $\kappa_\omega(k) = -\kappa_\omega(-k)$ and $\lambda_\omega(k) = \lambda_\omega(-k)$. Moreover, we shall see that all $\lambda_\omega$ are positive and display the equilibration rate of the mode $f_{k,\omega}$. The slowest equilibration is given by the smallest $\lambda_\omega$.

A general solution to (19) in terms of the eigenstates of $\mathfrak{D}_k$ is

$$\delta n(\theta, t, x) = \sum_k e^{ikx} \int d\omega \, c_{k,\omega} f_{k,\omega}(\theta) e^{-z_{k,\omega}t}, \tag{22}$$

where coefficients $c_{k,\omega}$ are specified by the initial state

$$\delta n(\theta, 0, x) = \sum_k e^{ikx} \int d\omega \, c_{k,\omega} f_{k,\omega}(\theta). \tag{23}$$

The integration is performed over $\omega$ parametrizing the spectrum of $\mathfrak{D}_k$ operator. In practice, when we perform numerical computations, operator $\mathfrak{D}_k$ becomes a finite matrix and its spectrum contains finite number of eigenvectors which turns the integration into a sum. The integration measure $d\omega$ is irrelevant, after discretization it can be absorbed into the coefficients $c_{k,\omega}$. Note that $\mathfrak{D}_k$ operator is not normal and therefore its eigenstates do not form on orthonormal basis. However, they do form a complete basis and (23) has a unique solution.

To quantify the time evolution we focus on the dynamics of momentum modes,

$$\delta n_k(t) = \int d\theta \, d\omega \, c_{k,\omega} f_{k,\omega}(\theta) e^{-z_{k,\omega}t}, \tag{24}$$

and on the moments of the filling function $\delta n(\theta, t, x)$,

$$\mu_j(t, x) = \int d\theta \, \theta^j \delta n(\theta, t, x). \tag{25}$$

We will also look at the diffusion coefficient describing the linear in time growth of the spatial variance of the particles' distribution. To this end, we define the diffusion coefficient $D_j$ associated to the $j$-th moment as

$$\mathrm{Var}(\mu_j)(t) \sim 2t D_j, \tag{26}$$

where

$$\mathrm{Var}(\mu_j)(t) = \int dx \left(x - \bar{x}_j\right)^2 \mu_j(t, x). \tag{27}$$

With $\bar{x}_j$ we denote the average position with respect to the $j$-th moment, defined as

$$\bar{x}_j = \int dx \, x \, \mu_j(t, x). \tag{28}$$

The relation (26) can hold only approximately. The time evolution, even in the linearized regime, is quite involved, and can not be simply described by a number of diffusion coefficients. However, we will see that for intermediate times the relation (26) holds and helps in clarifying the quantum fluid dynamics. At larger times it breaks due to the finite length $L$ of the system.

## 3.2 The diffusion operators

We will now display some properties of the diffusion operator $\tilde{\mathbf{D}}$ (5) and a related operator $\mathfrak{D}_k$ (20). The eigenvalues of $\tilde{\mathbf{D}}$ are non-negative real numbers as follows from its construction [43]. Examples of spectra of $\tilde{\mathbf{D}}$ obtained through numerical diagonalization for thermal and BEC-quench distributions are shown in Fig. 3. We observe that all eigenvalues of $\tilde{\mathbf{D}}$ but one depend continuously on the eigenvalue index. The special zero eigenvalue is related to the Markov property of the diffusion matrix [43]. The corresponding left eigenfunction is, in

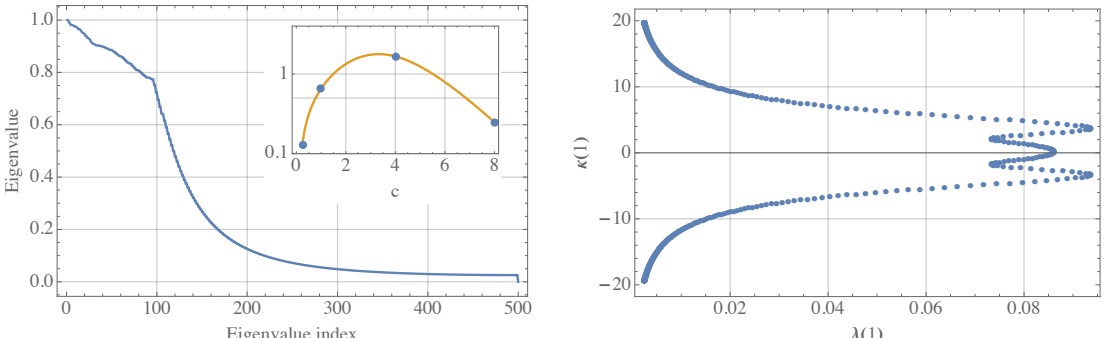

Figure 3: Spectra of the diffusion operators for a thermal state with $c = 1$ and $T = 1$. Left panel: Distribution of 500 eigenvalues of $\tilde{\mathbf{D}}$. The eigenvalues are normalized to the largest eigenvalue (about 0.19) and ordered from maximal to minimal. The cusps are the artefact of this ordering and appear when the degeneracy of the eigenvalues changes. The inset shows the dependence of the trace of $\tilde{\mathbf{D}}$ on the interaction strength in the BEC-quench saddle-point state. Right panel: Distribution of 500 eigenvalues $z_{k,\omega}$ of $\mathfrak{D}_k$ for $k = 1$. The eigenvalues form a complex spectral curve. Each point plotted correspond to a single eigenstate and eigenstates can be labeled uniquely by its imaginary part $\kappa(1)$.

the Lieb-Liniger model, the constant function. This can be easily seen from the definition of the diffusion kernel (4).

The Lieb-Liniger model in the $c \to 0$ limit is a theory of free bosons, whereas for $c \to \infty$ maps to free fermions (Tonks-Girardeau gas). In both cases, there is no diffusion because there is no interaction between quasi-particles. This is supported by the large $c$ analysis of the diffusion operator presented in [43]. In the inset of Fig. 3 we plot the dependence of the trace of $\tilde{\mathbf{D}}$ for the BEC-quench saddle point state as a function of the interaction strength $c$ which shows the expected behaviour.

We turn now to the analysis of the spectrum of the $\mathfrak{D}_k$ operator, see also Fig. 3. The diagonal operator with values $v^{\text{eff}}$ reigning the ballistic evolution and $\tilde{\mathbf{D}}$ do not commute with each other. This rebuilds the spectrum of the resulting operator $\mathfrak{D}_k$ with respect to that of $\tilde{\mathbf{D}}$. One of the consequences is that $\mathfrak{D}_k$ has no zero modes. Numerical computations reveal that the eigenvalues are smooth functions of $k$ and with great accuracy follow a scaling with $k$,

$$\kappa_\omega(k) \approx k\kappa_\omega(1), \qquad \lambda_\omega(k) \approx k^2\lambda_\omega(1), \tag{29}$$

as illustrated in Fig. 4. This simple scaling implies that modes with larger $k$ diffuse much faster than modes with smaller $k$. The eigenvalues $\kappa_\omega(k)$ describe the angular frequency of ballistic propagation of a mode, whereas eigenvalues $\lambda_\omega(k)$ are responsible for its diffusive decay. The dimensionless ratio of the numbers defines two regimes of the propagation where either ballistic or diffusive propagation dominates. Because of the observed scaling, we can introduce an effective momentum $k_\omega^*$

$$k_\omega^* = \frac{\kappa_\omega(1)}{\lambda_\omega(1)}. \tag{30}$$

Modes with $k > k_\omega^*$ propagate diffusively, modes with $k < k_\omega^*$ propagate ballistically. The effective momentum depends on the eigenstate and changes over few orders of magnitude as also illustrated in Fig. 4. The scaling of the eigenvalues is due to the fact that the diffusion is a small correction to the ballistic motion and would break if the diffusion was stronger. The strongly diffusive modes are however short-lived and do not influence the long-time relaxation

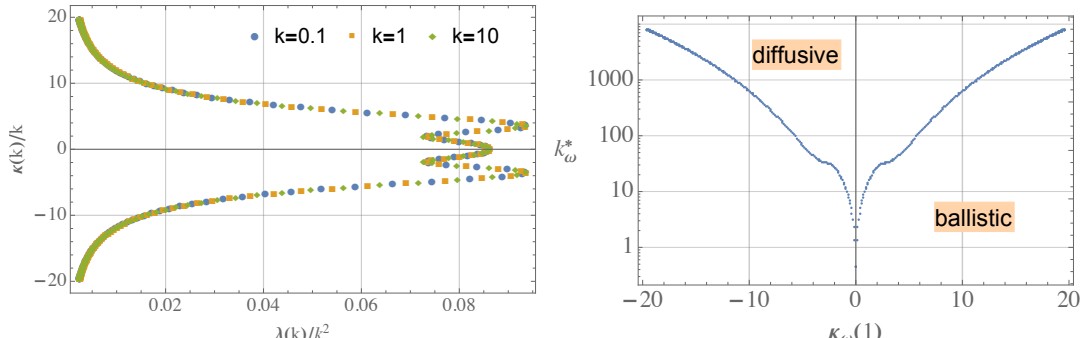

Figure 4: Left panel: We plot every third eigenvalue of $\mathfrak{D}_k$ operator for $k = 0.1, 1, 10$. In rescaled variables $(\lambda(k)/k^2, \kappa(k)/k)$ the spectral curves fall on each other exhibiting relation (29). Right panel: Dependence of the effective momentum $k_\omega^*$ on the eigenstates of the $\mathfrak{D}_k$ operator. We label the eigenstates by the (rescaled) imaginary part of the eigenvalue $z_{k,\omega}$. The diffusion has a small effect for most modes and therefore the ballistic propagation dominates even for relatively large momenta. The background state is the thermal state at $c = 1$ and $T = 1$.

mechanisms. Therefore, the scaling of eigenvalues and a perturbative picture of diffusion are universal at large enough times. This suggests that it might suffice to treat diffusion perturbatively to capture its effect.

# 4 Examples

In this section we consider the dynamics governed by the linearized GHD over a thermal state. We have checked that the dynamics over the BEC-quench saddle point state (12) yields qualitatively similar results. We consider two types of the initial state. The first one has a well-defined momentum and therefore its time evolution is especially simple. The second perturbation starts instead localized in space. The two examples serve as a good illustration of equilibration of the Lieb-Liniger fluid within the GHD. From now on we fix the background state to be the thermal equilibrium state of $c = 1$, $T = 1$ and unit density of particles.

## 4.1 Single Fourier mode initial state

We start with the single mode perturbation $\delta n$ of a well-defined momentum $k$. The initial perturbation is of the product form [3]

$$\delta n(\theta, 0, x) = \epsilon \cos(kx) \cdot \underline{n}(\theta)(1 - \underline{n}(\theta)), \tag{31}$$

where $\underline{n}$ is the GGE background filling fraction, see Fig. 5. The profile guarantees that the perturbation is small compared to $\underline{n}$ for all $\theta$ and all times.

For a perturbation with a well-defined momentum $k$ (with $k = 1 \, [k_F]$ here[4]), the equilibration follows a straightforward way. The dynamics cause a decay of the momentum mode as shown in Fig. 5. The effect of inclusion of the diffusion to the hydrodynamics leads to slightly faster decay. Therefore the main effect is played by the different velocities of propagation of particles with different rapidities. The subleading role of the diffusion confirms the observation from the previous section. Specifically, from right panel of Fig. 3 we can read off that

---

[3]In the following we shall suppress an overall small parameter $\epsilon$ because it scales out from the linear equation.
[4]The Fermi momentum $k_F = \pi$ for the unit density of the background state

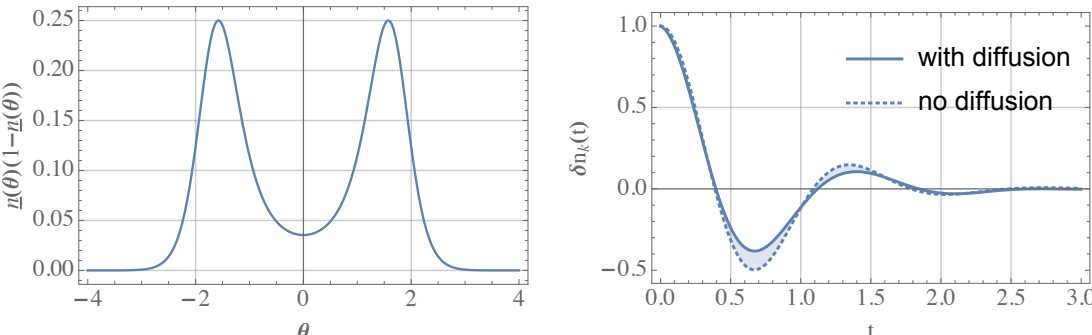

Figure 5: Left panel: the initial profile $\underline{n}(\theta)(1-\underline{n}(\theta))$. Right panel: the evolution of mode occupation $\overline{\delta n_{k=1}(t)}$ with and without the diffusion.

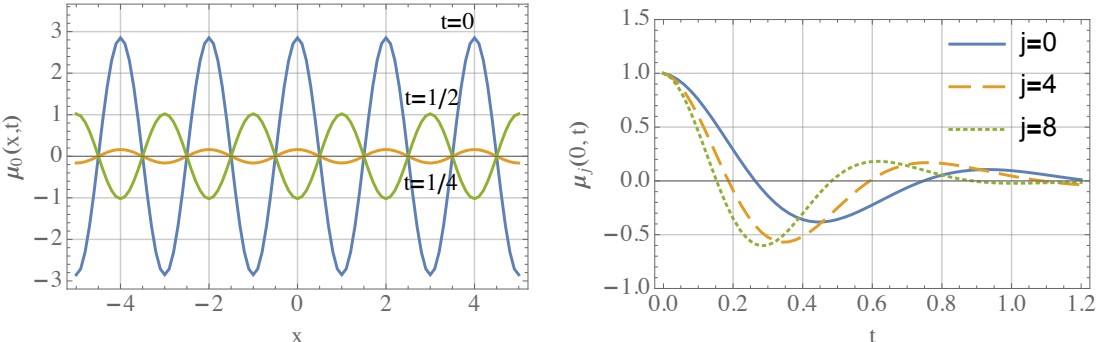

Figure 6: Time evolution for the initial profile (23). Left panel: the zeroth moment $\mu_0(t,x)$ shows steady collapse of the initial spatial profile. Right panel: local moments of the distribution $\mu_j(x=0,t)$ cease to zero with the $j=0$ moment being the slowest.

$\lambda_\omega(1) \sim 10^{-2}$ and is two-three orders smaller than $\kappa_\omega(1)$. In short, the ballistic propagation dominates. The time scale $t_d$ associated with the ballistic, dispersive, propagation is related to the characteristic scale $\kappa_\omega$ as $k_\omega t_d \sim 2\pi$. For the example discussed here $\kappa_\omega \sim 10$ which yields equilibration time due to dispersion $t_d \sim 1$ in agreement with Figures 5 and 6.

Spatial time evolution is displayed in Fig. 6 where we plot the 0-th moment of $\delta n(\theta, t, x)$. Because of the wave-like form of the initial state the time evolution is periodic in space. Therefore, we focus on $x = 0$, which corresponds to one of the maxima of the initial state, and consider local moments of $\delta n(\theta, 0, t)$, that is $\mu_j(0, t)$. During the time evolution, the local conglomerates of particles at $x$ multiplicities of $k/\pi$ get smeared over the whole system. Particles with higher rapidities travel faster and therefore the higher moments of the distribution vanish faster, see also Fig. 6.

Finally, we look into the rapidities distribution at a specific position in space. We choose again $x = 0$. The evolution of $\delta n(\theta, 0, t)$ is shown in Fig. 7 for linearized GHD with and without the diffusion. We observe that, whereas the averaged distribution in both cases vanishes similarly, the mechanism is different. The presence of diffusion causes a steady decay of the perturbation whereas purely dispersive evolution leads to a superposition of decoherent waves. The time scale of the diffusive decay is hard to estimate because what is observed is the mixture of diffusive and dispersive effects.

The physical picture behind the scrambling in the space of rapidities relates back to fig. 2: diffusion leads to scattering processes between quasi-particles and the background which re-organises the quasi-particle distribution. This increases the total entropy of the fluid [43] as

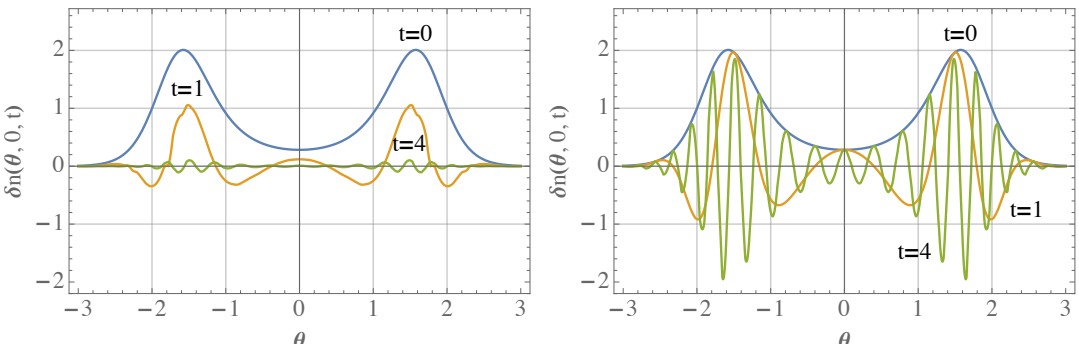

Figure 7: Time evolution of the local filling function $\delta n_k(\theta, 0, t)$ with (left panel) and without (right panel) the diffusion term in the linearized GHD.

we will now present.

For the Lieb-Liniger model the local entropy density, takes the familiar Yang-Yang form [50]

$$s(x,t) = -\int d\theta \rho_s(\theta, x, t)(n(\theta, x, t)\log n(\theta, x, t) + (1 - n(\theta, x, t))\log(1 - n(\theta, x, t)). \quad (32)$$

The total entropy $S(t)$ at time $t$ is then the spatial integral of $s(x,t)$ and according to the GHD predictions its production in time, in the leading order in the perturbation, is quadratic and given by [43]

$$\frac{\partial S(t)}{\partial t} = \frac{1}{2}\int dx \int d\theta \, d\alpha \frac{\partial_x \delta n(\theta, t, x)}{n(\theta)(1 - n(\theta))}\rho_s(\theta)\tilde{D}(\theta, \alpha)\partial_x \delta n(\alpha, t, x). \quad (33)$$

In purely non-diffusive GHD, with full non-linear dynamics, the entropy is conserved. Including the diffusion leads to the production of the entropy until the final state of the system settles in, as shown by the solid line in the right panel of fig. 7. When we linearize the GHD dynamics we introduce an error of the second order in the perturbation. This leads to the entropy production already at the non-diffusive level and to entropy production at the diffusive level larger then the one predicted by the full non-linear GHD with diffusion. These expectations are confirmed in the right panel of Fig. 7. The results show that scrambling in the space of rapidities is related to the entropy production.

We can summarize these findings concluding that the evolution of the perturbation is governed mainly by the dispersion suppression mechanisms. Particles with different rapidities propagate with different velocities which quickly (at the order of a single frequency period $\kappa_\omega(k)$) leads to equilibration of the initial spatially delocalized profile. However, looking into the details of local filling functions $\delta n(\theta, t, x)$ reveals the role of the diffusion in the equilibration. Whereas the ballistic propagation leads just to a decoherent superposition of waves, it is the diffusion that redistributes the rapidities and leads to a uniform, for a chosen initial state, long-time distribution.

## 4.2 Localized initial state

We consider now a bit more involved example of a perturbation localized initially in space. The initial configuration is

$$\delta n(\theta, t = 0, x) = e^{-\pi x^2}e^{-\theta^2}, \quad (34)$$

over the same thermal equilibrium background $\underline{n}(\theta)$. The time evolution and ultimately the equilibration, of the first and second moments $\mu_j(t, x)$, as defined in (25), is shown in Fig. 9.

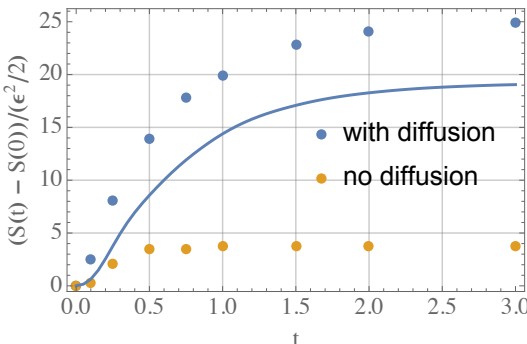

Figure 8: We plot the difference in the entropy between the state at time $t$ and the initial state. The dots corresponds to evaluating the entropy from the definition (32) with the help of the numerical solution considered here. The solid line is computed with the GHD prediction for the entropy production (33). The discrepancy between the GHD predictions and observed entropy originates from linearization of the full dynamics.

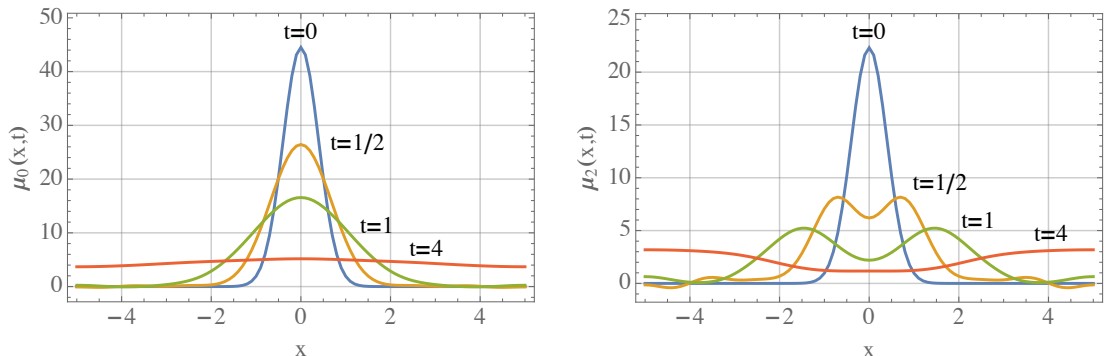

Figure 9: Dynamics of the initial Gaussian perturbation (34) over a thermal background with $c = 1$ and $\beta = 1$. Left panel: space-time dependence of the 0th moment $\mu_0(t, x)$ of the $\delta n(\theta, t, x)$ distribution. Right panel: plot of the 2nd second moment $\mu_2(t, x)$.

The final state of the system is given by the thermal background over which there is a uniform in space and time reminiscent of the initial perturbation given by (see (22))

$$\lim_{t \to \infty} \delta n(\theta, t, x) = \frac{2\pi}{L} \int d\omega \, c_{0,\omega} f_{0,\omega}(\theta). \tag{35}$$

It is worth pointing out that there are no dissipation processes and that the total density of quasi-particles is conserved

$$\int dx \, \delta n(\theta, t, x) = 2\pi \int d\omega \, c_{0,\omega} f_{0,\omega}(\theta). \tag{36}$$

At large times the density of quasi-particles gets spread uniformly over the space as (35) indicates.

To quantify the process of equilibration we consider the evolution of modes $\delta n_k(t)$ defined in (24). The results are presented in Fig. 10 and show that the larger momentum the faster the decay and that the diffusion slightly speeds up this process. We also estimate the diffusion coefficient for different moments $\mu_j(t, x)$, see also Fig. 9. The higher moments diffuse faster with the density (0-th mode) decaying the slowest.

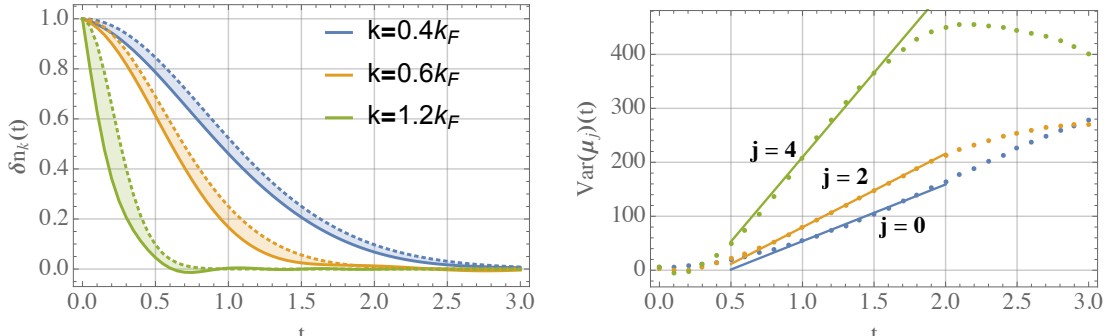

Figure 10: Left panel: the decay of different momenta modes $\delta n_k(t)$. The dotted line corresponds to non-diffusive hydrodynamics. Right panel: The time dependence of the variance of the position. In the linear regime we fit the diffusion coefficient according to (26). At late time the turn over of the variance is caused by the periodic boundary conditions.

Finally, we look at the "occupation numbers" $c_{k,\omega}(\theta,t)$ quantifying the number of ballistic and diffusive modes. We define the "ballistic occupation" and "diffusive occupation" by

$$c_{\text{ballistic}}(t) = \int d\omega \sum_{|k|<k_\omega^*} |c_{k,\omega}(t)|, \tag{37}$$

$$c_{\text{diffusive}}(t) = \int d\omega \sum_{|k|\geq k_\omega^*} |c_{k,\omega}(t)|, \tag{38}$$

with $k_\omega^*$ defined in (30). We expect the diffusive part to decay much faster than the ballistic part. These intuitions are confirmed on Fig. 11 where we show the time evolution of both quantities for the initial perturbation given by the Gaussian packet (34).

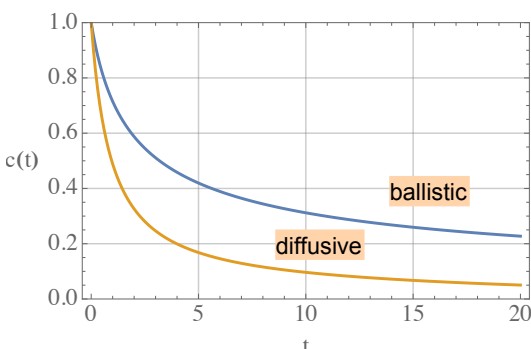

Figure 11: The time evolution of the ballistic and diffusive "occupation numbers" $c_{\text{ballistic}}(t)$ and $c_{\text{diffusive}}(t)$ from equations (37) and (38).

Concluding, the time evolution of the localized perturbation is again driven by the ballistic propagation with diffusion playing a secondary role. Moreover, the modes with diffusive dynamics decay faster. Despite this, there is a region in time in which the dynamics is diffusive, in the sense that the spatial variance of expectation values of local conserved charges grows linearly in time. This is caused by an interplay of the ballistic propagation, which mixes quasiparticles in the real space, and diffusive propagation, which mixes modes in the rapidities space.

# 5 Quadratic corrections

In this section we go one step beyond the linear approximation discussed until now and consider quadratic GHD. We focus on the k=0 mode as this mode does not evolve at the linear level. We also dismiss the quadratic part of the diffusion operator as it gives only subleading corrections to $k \neq 0$ modes.

The full GHD equation (15) can be easily expanded to the second order in $\delta n(\theta, t, x)$ keeping small diffusion term at linear order only:

$$\partial_t \delta n + v_{\underline{n}}^{\text{eff}} \partial_x \delta n + \delta v \, \partial_x \delta n = \frac{1}{2} \tilde{\mathbf{D}}_{\underline{n}} \partial_x^2 \delta n. \tag{39}$$

The leading second order correction comes from the effective velocity term because the latter depends on the full state $n(\theta, t, x)$. Directly from the definition of the dressing procedure in Appendix A we get

$$\delta v = v^{\text{eff}} \left( \frac{\delta E'}{E'} - \frac{\delta p'}{p'} \right) = \frac{(\mathbf{T} \delta n(E')^{\text{dr}})^{\text{dr}}}{(p')^{\text{dr}}} - v^{\text{eff}} \frac{(\mathbf{T} \delta n(p')^{\text{dr}})^{\text{dr}}}{(p')^{\text{dr}}}. \tag{40}$$

Expanding $\delta n$ and $\delta v$ in spatial Fourier modes leads to the following equation for the $k = 0$ mode

$$\delta n_0(\theta, t) - \int \mathrm{d}t \sum_{p+q=0} iq \, \delta v_p(\theta, t) \, \delta n_q(\theta, t) = 0. \tag{41}$$

Here, $\delta n_q$ is the linear solution to the GHD and determines $\delta v_p$ according to (40). The equation (41) can be now solved numerically with the techniques introduced in the previous sections.

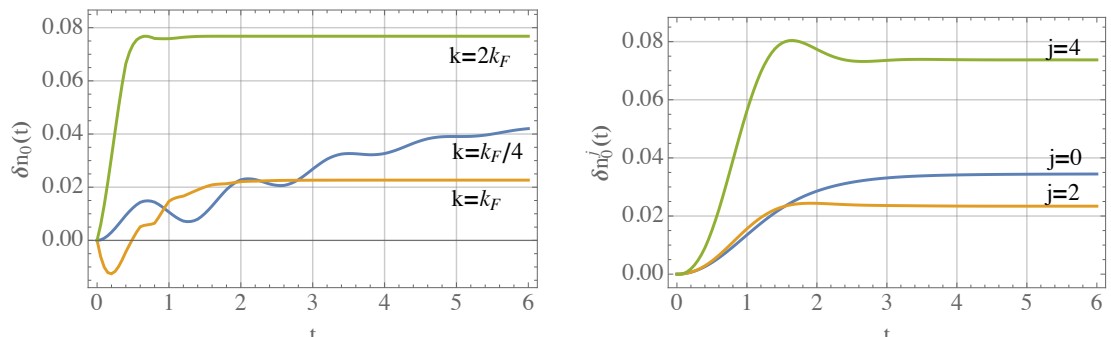

Figure 12: Left panel: The dynamics of the $k = 0$ mode following an initial state (31) with $\epsilon = 1$ and three different values of the momentum $k$. Right panel: The dynamics of the $k = 0$ mode following an initial Gaussian state of (34). Additionally to the density moments ($j = 0$), we plot the the second and fourth moments. In computing the time evolution we took into account only the smallest momentum modes of $\delta n_k(\theta, t, x)$.

In Fig. 12 we plot the dynamics of the $k = 0$ mode resulting from (41). In the left panel of Fig. 12 we consider the delocalized initial state from Eq. (31). Such a state has a well-defined momentum $k$ and we consider its three different values. The slower the momentum the slower the resulting evolution of the $k = 0$ mode to its late-time value.

We consider also the dynamics of the $k = 0$ mode with the localized initial state (34). To simplify the problem we approximate the full dynamics given by (41) by including only the

lowest contributing modes. This provides a lower bound on the equilibration process. In the right panel of Fig. 12, we plot the moments of the resulting distribution of quasi-particles,

$$\delta n_0^j(t) = \int d\theta \, \theta^j \delta n_0(\theta, t). \tag{42}$$

The higher moments, which depend more strongly on large $\theta$ part of the distribution $\delta n_0(\theta, t)$ equilibrate faster. This is simply caused by their larger effective velocity.

It is worth to stress that the $k = 0$ mode tends to a constant non-zero value as times goes to infinity. That means that quadratic fluctuations modify the equilibrium state. Whereas linear dynamics describes decaying fluctuations around the initial profile, inclusion of the quadratic terms leads to the change in the homogeneous profile itself. This is expected from general properties of GHD which has now quite solid foundation due to various numerical checks (see e.g. recent works [54, 55]). Here we can see it in a much simpler setup.

# 6 Conclusions

In this work, we have considered numerical solutions to the Generalized Hydrodynamics in the linear regime. This required turning an initial integro-differential equation into an eigen-problem of the diffusion operator. The latter being exactly known in terms of the quantities of the Thermodynamic Bethe Ansatz. We have performed numerical diagonalization of this operator. Questions about the analytic construction of its eigenstates remain open.

The general solution to the linearized GHD presented in Section 3 was then applied to two concrete examples of initially delocalized and localized states. In Section 4 we focused on the time evolution over the thermal background, however, we have checked that time evolution over the non-equilibrium saddle-point state like the one obtained after the BEC-quench leads to qualitatively similar results. The results show the complementary role of the ballistic and diffusion effect on the dynamics. Whereas the ballistic propagation dominates the spatial homogenization of the initial state, it is the diffusive dynamics that reorganizes the quasi-particle content and ultimately equilibrates distribution also in the rapidities space. It also increases the total entropy of the fluid but the linear approximation applied in this paper overestimates the exact result for the entropy production.

On the other hand, we have observed that the effect of the diffusion is rather weak and potentially can be treated in a perturbative way. A first step in developing such an approach, in a context of the bi-partite quench, was performed in [43]. It would be interesting to pursue this approach further.

Finally, we have also considered the effect of the quadratic terms. These are most important for the evolution of the $k = 0$ spatial mode which does not evolve in the linear regime. Quadratic corrections do not show any sign of instabilities which could appear due to positive interference of linear modes. Instead, they tend to non-zero values at late times modifying the homogeneous GGE state. This behavior supports the GHD as a valid approach to non-equilibrium physics of the integrable systems.

In this work we focused on the simplest and most easily accessible observables. It would be interesting to consider the dynamics of other observables, e.g. the local $n$-body correlations functions for which exact expressions in the homogeneous Lieb-Liniger model were recently obtained [56]. Under the hydrodynamic assumption, these can be extended to the inhomogeneous case and evaluated for the linearized or full GDH dynamics. We leave this problem for future work.

## Acknowledgments

MP thanks Jacopo De Nardis and Piotr Szymczak for insightful discussions and acknowledges the support from the National Science Centre under SONATA grant 2018/31/D/ST3/03588 at the final stages of this work.

## A    Integral operators

In the main text we use a shorthand notation for the action of integral operators. For any $\mathbf{K}(\theta, \theta')$ its action on a function $f(\theta)$ is denoted and understood as

$$\mathbf{K}f \longrightarrow \int \mathrm{d}\theta' \mathbf{K}(\theta, \theta') f(\theta'). \tag{43}$$

The dressing procedure is defined through the following integral equation

$$f^{\mathrm{dr}}(\theta) = f(\theta) + \int \mathrm{d}\theta' T(\theta, \theta') n(\theta') f^{\mathrm{dr}}(\theta'). \tag{44}$$

Here $T(\theta, \theta')$ is the differential scattering kernel and $n(\theta)$ is the filling function. In the compact notation introduced above it is written as

$$f^{\mathrm{dr}} = f + \mathbf{T} n f^{\mathrm{dr}}. \tag{45}$$

Introducing the resolvent $(1 - \mathbf{T}n)^{-1}$ of the kernel $\mathbf{T}n$ the solution to this equation takes the form

$$f^{\mathrm{dr}} = (1 - \mathbf{T}n)^{-1} f. \tag{46}$$

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
