# Peer review of "Linearized regime of the generalized hydrodynamics with diffusion"

_SciPost Physics, doi:SciPost Phys. Core 1, 002 (2019)_

## Round 2 · Referee Report · Anonymous · 2019-7-25

Strengths

1- the numerical results are convincing
2- interesting discussion of eigenvalues of diffusion operator and of separation between ballistic and diffusive modes
3- some beyond-linear effects studied

Weaknesses

1- some inaccuracies in the explanations
2- lack of clarity, in my opinion, in the definition of some important quantities
3- mainly numerical, analytics is limited
4- the analysis stays somewhat shalow

Report

In this paper, the authors study the linearised regime of the equations of generalised hydrodynamics with diffusion. They identify the modes dominated by ballistic propagation, and those by diffusive. The study is mainly numerical, with various initial conditions considered.

I think the paper is a fine study of diffusive effects in generalised hydrodynamics. However, there are many inaccuracies in the explanations, and especially in the definition of $\omega(k)$ and in relations involving it. I think this should be greatly improved before the paper can be published in Scipost Physics.

Clarifications in the general GHD explanations:

- Page 3: “Locally, the densities … Gibbs ensemble” not clear - with diffusion, operators are not locally given by GGE averages, as there are derivative corrections.

- Page 3: note that without diffusion, [31] presents a solution framework for (1) by integral equations

- Page 3: in fact no shocks can be sustained in GHD, i.e. large spatial variations may appear but they then always disappear.

- Page 3, after eq 2: $(E’)^{dr}$ is used, and afterwards $E^{dr}$ is discussed, this is not so clear. Note: the derivative cannot be brought outside of dressing in general. In fact, it is $\int d \theta (E’)^{dr}$ that represents the energy of the particle above a bath (sim. for momentum). There is another “dressing operation”, say Dr, which would be such that $E^{Dr}$ is the correct energy above a bath. See e.g. discussions in [43] and references cited there

- Page 3: mention what ${\bf T}$ is (differential scattering kernel), refer to later equation.

- Page 5: (15) is correct at large times only if disturbance of initial state, which may be very strong, is nevertheless compactly supported; if it is extended, then there is no reason for (15) to hold, for instance in the partitioning protocol.

Clarifications in the main results / arguments:

- Page 7: I don’t understand eq 22. $\omega$ labels the eigenvalues, but how is it a function of $k$? For a given $k$ there are infinitely many values of $\omega$, I don’t see how it becomes a function of $k$.

- Relation 27 approximate in what sense? At large times?

- Page 8: I don’t understand the scaling law. My understanding is that given a $k$, there are eigenvalues $\omega$ for $\tilde{\bf D}_k$. These have a real and imaginary part. $\kappa_\omega$ is the real part. There are many such eigenvalues, in the limit forming a continuum supposedly, and this, for every $k$. What does the dependence on $k$ mean? What does it mean that they scale in this way? As you change $k$, how do you fix the next eigenvalue you look at in order to determine how it changes with $k$? Or do you simply mean that the range of values taken by real / imaginary parts grows like $k$ / $k^2$? Is this scaling law valid at large $k$, or approximately for all values of $k$? Please do provide a graph that shows clearly the scaling, and in what sense.

- Page 8,9: effective momentum depends on eigenstate: what eigenstate? Should I choose $k=1$, then choose a $\omega$ (hence an eigenstate) and calculate the ratio between real and imaginary part? Is the eigenstate dependence really the $\omega$ dependence? What is the “eigenstate index”? How do you order the eigenstates?

- Pages 9 and 11, eqs 32,33: what is the small parameter here for the linear approximation? In the numerics, there should be a small parameter introduced as otherwise the equations are not quite correct; or at least one should justify the expected strength of the non-linear effects.

- Section 5: how does the quadratic part of the expansion on other modes interplay with that of the zero-mode? How can we justify then to concentrate on the zero mode?

Small things:

- Page 8: is it really ${\bf D}$ or rather the more natural $\tilde{\bf D}$ whose eigenvalues are evaluated?

- Page 8: “We observe that all eigenvalues but one are of the same order ” you mean all fall onto some kind of continuous functions except one; “same order” is a bit unclear.

- Page 8: $\tilde D_k$ instead of $n$?

- Pages 11,12: is $C_{k,\omega}$ the same as $c_{k,\omega}$ used in previous pages?

Requested changes

Clarifications as per the points made in the report

  • validity: good
  • significance: ok
  • originality: ok
  • clarity: low
  • formatting: excellent
  • grammar: good

Author:  Milosz Panfil  on 2019-09-13  [id 603]

(in reply to Report 1 on 2019-07-25)
Category:
answer to question

We would like to thank the referee for the report and insightful comments. We made substantial changes to the manuscript to comply with the issues raised by the referee, the full list of changes accompanies the resubmission. We agree with the referee on all the points raised and improved the presentation. Specifically

1) Clarifications in the general GHD explanations:

We followed the referee's comments and suggestion for the improvements

2) Clarifications in the main results / arguments:

  • Regarding the spectrum of $\tilde{D}_k$

First point to note is that the spectrum is non-degenerate and therefore the eigenvectors can be uniquely labelled by their eigenvalues $\omega(k)$. When we vary $k$ we find different eigenstates and different eigenvalues. The eigenvalues are complex and form a line, spectral curve in the complex plane. When we plot them, we observe that the spectral curves for different $k$ follow the simple scaling with $k$/$k^2$. In a new version, we have provided a plot that illustrates this scaling.

The scaling appears because the diffusion has a small effect on the spectrum. We don't see any signs of its breaking even for $k$ of order of hundreds.

  • Page 8,9: effective momentum depends on eigenstate: what eigenstate?

The eigenstates are labelled by $\omega(k)$, specifying $\omega(k)$ specifies the eigenstate. Actually, it is enough to look at the imaginary part of $\omega(k)$, $\kappa(k)$. We have remade the plot, fig. 4, changing the ambiguous “eigenstate index” with the imaginary value of \omega.

  • Pages 9 and 11, eqs 32,33: what is the small parameter here for the linear approximation?

We have introduced a small parameter $\epsilon$ that makes the perturbation small with respect to the background state. It is worth pointing out, however, that this parameter drops from the linearized equations for the dynamics of the perturbation.

  • Section 5: how does the quadratic part of the expansion on other modes interplay with that of the zero-mode? How can we justify then to concentrate on the zero mode?

The leading order of the zeroth mode is given by the linear evolution of the $k \neq 0$ modes, see eq. (41). Including the second order dynamics of $k \neq 0$ modes in that equation would modify the subleading, not leading, terms of the zeroth mode dynamic’s.

We concentrate on the zero mode dynamics because all other modes propagate already at the linear level and thus the second order gives only subleading corrections to them.

3) Small things

We thank the referee for pointing out all the typos and inconsistencies. We have corrected them.

---

## Round 2 · Referee Report · Anonymous · 2019-7-31

Strengths

- timely subject
- interesting observations on non-equilibrium dynamics and scrambling of quasi-particles

Weaknesses

- physical consequences of the scrambling of quasi-particles content are not presented

Report

The author study the presence of diffusive spreading in the generalised hydrodynamic setting for integrable model (here the Lieb-Liniger model). They show that relaxation in real space is mostly dominated by ballistic spreading while relaxation in the quasi-particle space by diffusive spreading. Diffusion terms therefore are the main scrambler of quasiparticle dynamics, see Fig. 7.

It would be good to have a more clear physical picture of such scrambling. For example can this be seen by entropy increase due to diffusion? It was noticed in https://journals.aps.org/prl/abstract/10.1103/PhysRevLett.120.164101 that during the non-equilibrium dynamics of an hard rod gas with no diffusion there is formation of fractal structure in rapidity space. By coarse graining over them one obtain a new coarse-grained Euler scale evolution with larger entropy. The entropy increase was compared with formulas for hard rods in the supplemental materials of that paper. Now the entropy increase is known also for a quantum gas, see https://scipost.org/10.21468/SciPostPhys.6.4.049 so it would be good to compare with this expression.

If the two evolution, diffusive and not, in Fig 7 are so different I would expect that operators that are less and less locals display different relaxations in time. Now that all moments of density fluctuations in the LL gas are known, https://journals.aps.org/prl/abstract/10.1103/PhysRevLett.120.190601, I would suggest to display the dynamics of larger moments and compare with the evolution with diffusive terms, can we then see a more pronounced difference for the higher moments?

The authors should comment the origin of the cusp in Fig 3, left, and why the maximal eigenvalue does not vanish as c->0 when indeed the theory should become free.

Page 8 "due to the non-vanishing commutator" the authors should explain what they mean by that.

Page 14 : "this means that quadratic fluctuations modify the equilibrium GGE ..." the authors should explain what they mean by that.

Change quasiparticles with quasi-particles or the reversed consistently.

Requested changes

- quantify the scrambling in rapidity space via entropy increase
- check the evolution of powers of the density operator in time
- make statements more clear

  • validity: high
  • significance: high
  • originality: good
  • clarity: good
  • formatting: good
  • grammar: excellent

Author:  Milosz Panfil  on 2019-09-13  [id 602]

(in reply to Report 2 on 2019-07-31)
Category:
answer to question

We would like to thank the referee for the report and for raising interesting questions.

1) quantify the scrambling in rapidity space via entropy increase

That’s a great idea, and indeed, the entropy production is related to the scrambling in the rapidities space. We have added a discussion and a plot in that respect at the end of Section 4.1. Note that, at the linear level, the predictions for the behaviour of the entropy are a bit different from the results of full dynamics. Specifically, even without the diffusion the entropy is produced, further more inclusion of diffusion leads to larger entropy production than the full GHD predicts. We added a discussion of these issues in the main text.

2) check the evolution of powers of the density operator in time

We agree with the referee that it would be interesting to study the dynamics of more involved observables, however we feel that that would really required substantial extension of our present work. Given our focus on understanding how diffusion works at the level of linearized hydrodynamics, the dynamics of local n-body correlations functions is beyond our present scope. Still, an interesting and relevant question for future work.

3) the origin of the cusp in Fig 3, left, and why the maximal eigenvalue does not vanish as c->0

The cusp is the artefact of the ordering of the eigenvalues of $\tilde{D}$. The cusps appear when the degeneracy of the eigenvalues changes first from 4-fold degeneracy to 6-fold degeneracy and then to 2-fold degeneracy.

With respect to the second point, the maximal eigenvalue is not the best quantity to measure the strength of the operator. Therefore we have changed the plot and display now the trace of $\tilde{D}$. The apparent raising of the maximal eigenvalue at the previous plot was due to instabilities of numerical diagonalization appearing when the interaction parameter $c$ was very small.

4) Regarding the other points, we have improved the presentation and made the notation consistent.

---

## Round 3 · Referee Report · Anonymous (Referee 3) · 2019-9-17

Report

The authors implemented all the points I raised and I believe the paper can now be published as it stands.

---

## Round 3 · Referee Report · Anonymous (Referee 4) · 2019-10-15

Report

I thank the authors for the changes they have made.

The introduction is now clearer, as is most of the discussion.

I now understand what they mean by ω(k), which has been clarified with figure 4. However, some equations still don’t make sense. The problem is that the authors take the intuition from the case where the k-dependent operators all commute with each other, with a common set of eigenfunctions a different eigenvalues - in this case omega(k) makes sense, as a given eigenfunction can be identified for all operators, with a k-dependent eigenvalue. Here this is not the case. Each operator has its own set of eigenfunctions, and set of eigenvalues. So there is not a unique a priori” way of getting a *function of k* ω(k) for the eigenvalues - a priori k-independent way of identifying a given eigenvalue. But, from figure 4, what they have done is simply counting the eigenvalue from that with smallest imaginary part to that with largest. This is fine, and indeed gives ω(k) and one can analyse the scaling.

However, equations (22) and (24) still don’t make sense without further explanations. ω appears as an integration variable - over the reals? - yet it appears in the exponential as a function of k.

Please adjust equations (22) and (24) or add explanations around, and also clarify from the beginning, from eq (21), what you mean by ω(k) (e.g. its not “the eigenvalue of k”, because there isn’t a unique eigenvalue, there is rather a continuum).

Once these small points are assessed, the paper can be published.

  • validity: -
  • significance: -
  • originality: -
  • clarity: -
  • formatting: -
  • grammar: -

Author:  Milosz Panfil  on 2019-11-04  [id 638]

(in reply to Report 2 on 2019-10-15)
Category:
correction

Dear Referee,

Thank you again for the constructive comments.

We have clarified the presentation in the two places that you have mentioned.

---

## Round 3 · Author Response

Dear Editor,

Please find the new version of our manuscript for resubmission.
There is a number of modifications, additions and improvements that comply with the referees' reports.

Yours sincerely,
the authors

---

## Round 3 · List of Changes

The major changes are:

1) improved the general introduction to the GHD and Lieb-Liniger model.
2) changed the inset of Fig. 3, left panel. Instead of plotting the maximal eigenvalue we plot now the trace.
3) added a new figure, Fig. 4, left panel, showing the scaling of eigenvalues of Dk.
4) changed Fig. 4, right panel. Instead of ambiguous "Eigenstate index" we label the eigenstates by the imaginary part of the eigenvalues
5) added a discussion on entropy production and its relation to the equilibration in the rapidities space. There is also a new figure, Fig. 8.

We have also improved presentation in a number of places, including the ones suggested by the referees.

---

## Round 4 · Author Response

Dear Editor,

Please find the new version of our manuscript which clarifies the two points raised by the second referee.

Yours sincerely,
the authors

---

## Round 4 · List of Changes

1) we have changed the notation of the eigenvalues from $\omega(k)$ to $z_{k,\omega}$ to highlight that for each $k$ there is a continuum of eigenvalues labelled by $\omega$.

2) this change clarifies also the integrations in eqs. (22) and (24), on which we additionally comment below eq. (23).

---

## Editorial Decision

published